# Searching for an exotic spin-dependent interaction with a single electron-spin quantum sensor

Xing Rong [1,2,3], Mengqi Wang[1,2,3], Jianpei Geng[1,2], Xi Qin[1,2,3], Maosen Guo[1,3], Man Jiao[1,3], Yijin Xie[1,3], Pengfei Wang[1,2,3], Pu Huang[1,2,3], Fazhan Shi[1,2,3], Yi-Fu Cai[4,5], Chongwen Zou[6] & Jiangfeng Du[1,2,3]

Searching for new particles beyond the standard model is crucial for understanding several fundamental conundrums in physics and astrophysics. Several hypothetical particles can mediate exotic spin-dependent interactions between ordinary fermions, which enable laboratory searches via the detection of the interactions. Most laboratory searches utilize a macroscopic source and detector, thus allowing the detection of interactions with submillimeter force range and above. It remains a challenge to detect the interactions at shorter force ranges. Here we propose and demonstrate that a near-surface nitrogen-vacancy center in diamond can be utilized as a quantum sensor to detect the monopole–dipole interaction between an electron spin and nucleons. Our result sets a constraint for the electron–nucleon coupling, $g_s^N g_p^e$, with the force range 0.1–23 μm. The obtained upper bound of the coupling at 20 μm is $g_s^N g_p^e < 6.24 \times 10^{-15}$.

[1] CAS Key Laboratory of Microscale Magnetic Resonance and Department of Modern Physics, University of Science and Technology of China (USTC), Hefei 230026, China. [2] Hefei National Laboratory for Physical Sciences at the Microscale, USTC, Hefei 230026, China. [3] Synergetic Innovation Center of Quantum Information and Quantum Physics, USTC, Hefei 230026, China. [4] CAS Key Laboratory for Research in Galaxies and Cosmology, Department of Astronomy, USTC, Hefei 230026, China. [5] School of Astronomy and Space Science, USTC, Hefei 230026, China. [6] National Synchrotron Radiation Laboratory, USTC, Hefei 230026, China. Xing Rong, Mengqi Wang and Jianpei Geng contributed equally to this work. Correspondence and requests for materials should be addressed to P.W. (email: wpf@ustc.edu.cn) or to J.D. (email: djf@ustc.edu.cn)

Development of new techniques to search for new particles beyond the standard model is important in eliminating our ignorance of the ultraviolet completion of particle physics[1]. A type of hypothetical ultralight scalars, such as axions or axion-like particles (ALPs)[2], has attracted a lot of attention in a wide variety of researches. This has been well motivated for decades from the need of cosmology[3], namely, the dark matter candidate[4], the dark energy candidate[5], and from the understanding of the symmetries of charge conjugation and parity in quantum chromodynamics (QCD)[6] as well as predictions from fundamental theories such as string theory[1]. The exchange of such particles results in spin-dependent forces, which were originally investigated by Moody and Wilczek[7]. Various laboratory ALP searching experiments focus on the detection of macroscopic monopole–dipole forces between polarized electrons/nucleons and unpolarized nucleons[8–15]. Previous laboratory searching has set the limit on the monopole–dipole coupling between electron and nucleon, $g_s^N g_p^e$, with a force range $\lambda > 20\,\mu m$[16]. The experimental investigation of this interaction at force range shorter than 20 μm, however, remains elusive due to the following challenges: (i) the size of the sensor should be small compared to the micrometer force range; (ii) the geometry of the sensor should allow close proximity between the sensor and the source; (iii) the sensitivity of the sensor should be sufficient for searching or for providing stringent bound for such interaction; (iv) the unwanted noises, such as the magnetic and electric field introduced by environment, should be isolated well.

Here we develop a method to investigate the electron–nucleon monopole–dipole interactions using a near-surface electron-spin qubit in diamond. Constraints for the electron–nucleon coupling, $g_s^N g_p^e$, have been set for the interaction range 0.1–23 μm. For a force range of 20 μm, our constraint is bounded to be less than $6.24 \times 10^{-15}$. The method can be further extended to investigate other spin-dependent interactions[17] and opens the door for the single-spin quantum sensor to explore new physics beyond the standard model.

## Results

**Monopole–dipole interaction and experimental system.** We use a near-surface single electron spin, which is a nitrogen-vacancy (NV) center in diamond, to investigate the monopole–dipole interaction between an electron spin and nucleons. The axion-mediated monopole–dipole interaction can be described as[17]

$$V_{sp}(\mathbf{r}) = \frac{\hbar^2 g_s^N g_p^e}{8\pi m}\left(\frac{1}{\lambda r} + \frac{1}{r^2}\right)e^{-\frac{r}{\lambda}}\boldsymbol{\sigma}\cdot\mathbf{e}_r, \tag{1}$$

where $\mathbf{r}$ is the displacement vector between the electron and nucleon, $r = |\mathbf{r}|$ and $\mathbf{e}_r = \mathbf{r}/r$ are the displacement and the unit displacement vector, $g_s^N$ and $g_p^e$ are the scalar and pseudoscalar coupling constants of the ALP to the nucleon and to the electron, $m$ is mass of the electron, $\lambda = \hbar/(m_a c)$ is the force range, $m_a$ is the mass of the ALP, $\boldsymbol{\sigma}$ is the Pauli vector of the electron spin, $\hbar$ is Plank's constant divided by $2\pi$, and $c$ is the speed of light. Such interaction is equivalent to the Hamiltonian of the electron spin in an effective magnetic field $\mathbf{B}_{sp}(\mathbf{r})$ arising from the nucleon,

$$\mathbf{B}_{sp}(\mathbf{r}) = \frac{\hbar g_s^N g_p^e}{4\pi m\gamma}\left(\frac{1}{\lambda r} + \frac{1}{r^2}\right)e^{-\frac{r}{\lambda}}\mathbf{e}_r, \tag{2}$$

where $\gamma$ is the gyromagnetic ratio of the electron spin.

An NV-based optically detected magnetic resonance setup combined with an atomic force microscope (AFM) (shown in Fig. 1, see Supplementary Fig. 1 and Supplementary Note 1 for details) is constructed to search for this electron–nucleon interaction. A near-surface electron spin, which is a defect in diamond composed of a substitutional nitrogen atom and a neighboring vacancy[18], is utilized as a quantum sensor to detect its electron–nucleon interaction with nucleons in a fused silica half-ball lens. The NV center is <10 nm close to the surface of the diamond, so that it allows close proximity between the electron and the nucleon. Hereafter, the electron spin of the NV center and the half-ball lens are denoted as $S$ and $M$ for convenience, respectively. $M$ is placed on a tuning fork actuator of the AFM, which enables us to position $M$ near and away from $S$, as well as to drive $M$ to vibrate with a frequency. Figure 1b shows the geometric parameters in the experiment. The radius of $M$ is $R = 250(2.5)\,\mu m$. The vibration amplitude of $M$ is denoted as $A$. The time-dependent distance between the bottom of $M$ and $S$ can be described as $d = d_0 + A[1 + \cos(\omega_m t)]$, where $d_0$ is the minimal distance between $M$ and $S$, and $\omega_m$ is the vibration angular frequency of $M$ driven by the tuning fork. The effective magnetic field felt by $S$ arising from the hypothetic electron–nucleon interaction can be derived by integrating Eq. (2) over all the nucleons in $M$ as $\mathbf{B}_{eff} = \mathbf{e}_{r_c} B_{eff}$, where $\mathbf{e}_{r_c}$ is

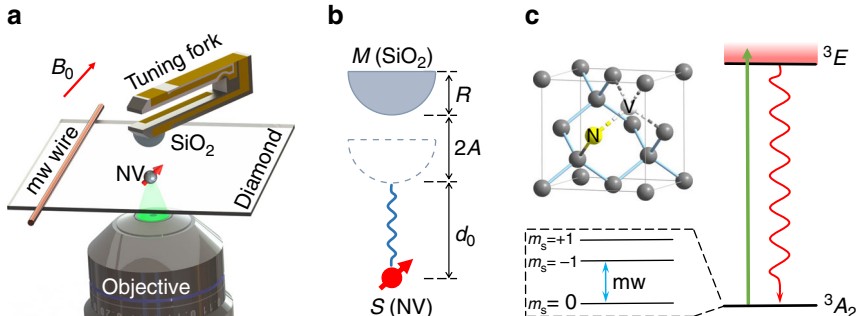

**Fig. 1** Experimental setup and the quantum sensor. **a** Schematic experimental setup. An NV center in diamond, which is labeled as NV, is used to search for the monopole–dipole interaction with nucleons. The nucleons are provided by a fused silica half-ball lens, which is labeled as SiO$_2$. The half-ball lens is placed on a tuning fork actuator of an AFM. A static magnetic field $\mathbf{B}_0$ is applied along the symmetry axis of the NV center. **b** Schematic experimental parameters. The electron spin and the half-ball lens are denoted as $S$ and $M$, respectively. The radius of $M$ is $R$. $M$ is located right above $S$ and driven to vibrate with amplitude $A$. The distance between $S$ and the bottom of $M$ is $d_0$ when $M$ vibrates to the position nearest $S$. **c** Atomic structure and energy levels of the NV center in diamond. The NV center consists of a substitutional nitrogen atom with an adjacent vacancy cite in the diamond crystal lattice. The ground and excited states are denoted as $^3A_2$ and $^3E$. The NV center can be excited from $^3A_2$ to $^3E$ by a laser pulse, and decays back to $^3A_2$ emitting photoluminescence. The optical transitions are used to initialize and readout the spin state of the NV center. The spin states $|m_S = 0\rangle$ and $|m_S = -1\rangle$ of $^3A_2$ are encoded as a quantum sensor. The state of $S$ can be manipulated by microwave pulses

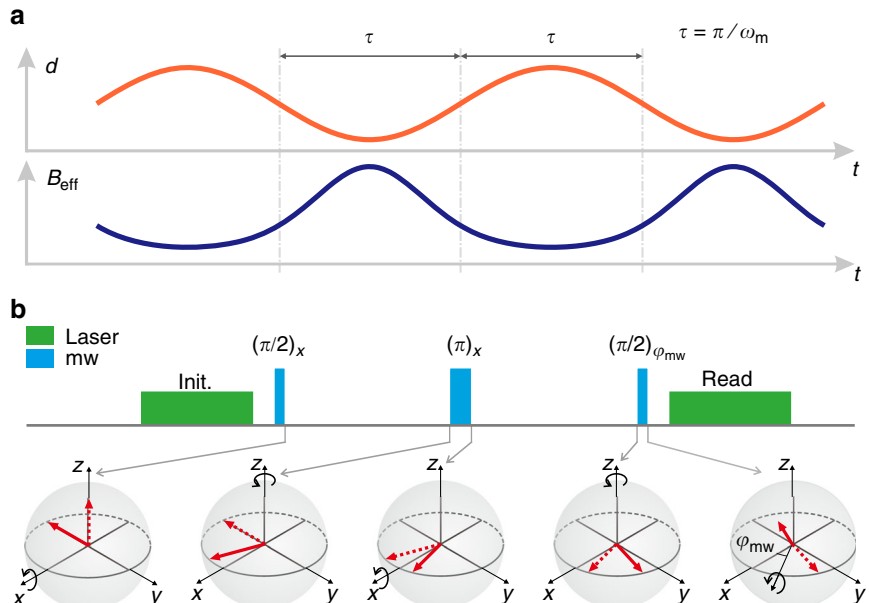

**Fig. 2** Electron–nucleon interaction detection scheme. **a** Time variation of the distance $d$ (upper) and the effective magnetic field $B_{eff}$ (lower). The distance $d$ is between $S$ and the bottom of $M$. The waiting time, $\tau = \pi/\omega_m$, is half period of the vibration of $M$, and $B_{eff}$ is the effective magnetic field on $S$ generated by the nucleons in $M$. **b** Experimental pulse sequence (upper) and state evolution of $S$ (lower). The pulse sequence applied on $S$ is synchronized with the vibration of $M$. Green laser pulses were used to initialize and read the state of $S$. The microwave $\pi/2$ and $\pi$ pulses were applied only when $M$ passed through the equilibrium point of the vibration

the unit distance vector along the symmetry axis of $M$ and

$$B_{eff} = \frac{\hbar g_s^N g_p^e \rho}{2m\gamma} f(\lambda, R, d),\qquad(3)$$

with $\rho = 1.33 \times 10^{30}~\text{m}^{-3}$ being the number density of nucleons in $M$ and $f(\lambda, R, d) = \lambda \left[ \frac{R}{d+R} e^{-\frac{d}{\lambda}} - e^{-\frac{d+R}{\lambda}} + e^{-\frac{\sqrt{R^2+(d+R)^2}}{\lambda}} + \frac{\lambda \sqrt{R^2+(d+R)^2}}{(d+R)^2} e^{-\frac{\sqrt{R^2+(d+R)^2}}{\lambda}} - \frac{\lambda d}{(d+R)^2} e^{-\frac{d}{\lambda}} + \frac{\lambda^2}{(d+R)^2} e^{-\frac{\sqrt{R^2+(d+R)^2}}{\lambda}} - \frac{\lambda^2}{(d+R)^2} e^{-\frac{d}{\lambda}} \right]$ (see Supplementary Note 2 for details). If $M$ is moved far away from $S$ with distance much larger than the force range $\lambda$, the monopole–dipole interaction is negligible. By comparing the magnetic field detected by $S$ with and without $M$, the electron–nucleon interaction between $S$ and the nucleons in $M$ can be measured.

Figure 1c shows the atomic structure and energy levels of the NV center. The ground state of the NV center is an electron-spin triplet state $^3A_2$ with three substates $|m_S = 0\rangle$ and $|m_S = \pm 1\rangle$. A static magnetic field $B_0$ of about 300 G is applied along the NV symmetry axis to remove the degeneracy of the $|m_S = \pm 1\rangle$ spin states. The spin states $|m_S = 0\rangle$ and $|m_S = -1\rangle$ are encoded as a quantum sensor[19]. Microwave pulses with frequency matching the transition between $|m_S = 0\rangle$ and $|m_S = -1\rangle$ are delivered by a copper microwave wire to manipulate the state of the quantum sensor. The $|m_S = 1\rangle$ state remains idle due to the large detuning. A laser pulse can be applied to pump the NV center from $^3A_2$ to the excited state $^3E$. When the NV center decays back to $^3A_2$, photoluminescence can be detected. The optical process can be utilized to realize state initialization and readout of this quantum sensor. Because of the convenient state initialization and readout procedures, precise control[20], long coherence time[21], and its atomic size, the NV center serves as a magnetic sensor at nanometer scale, which is now extended to search for the axion-mediated interactions beyond the standard model.

**Pulse sequence to detect the monopole–dipole interaction.** If mass $M$ is placed near the electron spin $S$, a static effective DC magnetic field $B_{eff}$ caused by monopole–dipole interaction will affect $S$. A straightforward approach to detect such DC magnetic field is to perform a Ramsey sequence[19]. The Ramsey sequence can be written as $\pi/2 - \tau - \pi/2$, where $\pi/2$ stands for the microwave pulse with rotating angle $\pi/2$ and $\tau$ stands for a waiting time. The first $\pi/2$ microwave pulse prepares $S$ to a superposition state $(|0\rangle - i|1\rangle)/\sqrt{2}$. During the waiting time $\tau$, the electron spin precesses about the $z$ axis and accumulate a phase proportional to the strength of the magnetic field $B_{eff}$. After the second $\pi/2$ pulse, the phase information will be encoded in the population of the state $|m_S = 0\rangle$, which can be detected with a laser pulse. However, during the waiting time, noises, such as the fluctuation of the Overhauser field and the slow drift of the external static magnetic field, will cause the dephasing. Thus the sensitivity of such method is limited by the dephasing time of the electron spin, which is about $T_2^* = 0.67(4)~\mu s$ measured in our experiment.

To suppress the dephasing and to enhance the sensitivity of detecting $B_{eff}$, a spin echo sequence[22] can be applied instead of the Ramsey sequence. The spin echo sequence can be written as $\pi/2 - \tau - \pi - \tau - \pi/2$, where $\pi/2$ ($\pi$) stands for the microwave pulse with rotating angle $\pi/2$ ($\pi$) and $\tau$ stands for a waiting time. With this spin echo sequence, the coherence time of the electron spin is enhanced to about $T_2 = 8.3(8)~\mu s$ in our experiment, which is of an order longer than $T_2^*$. Since the positive phase accumulated during the first waiting time $\tau$ is exactly canceled by the negative phase accumulated during the second $\tau$, the total phase due to static $B_{eff}$ is zero. To solve this problem, we drive $M$ to vibrate periodically to make $B_{eff}$ an oscillating signal (shown in Fig. 2a). If $B_{eff}$ is modulated in phase with the spin echo sequence, a nonzero accumulated phase due to $B_{eff}$ can be obtained, while the unwanted noise can be canceled. We use a homebuilt pulse generator and a comparator to make sure that the tuning fork oscillation and the pulse sequence are synchronized well (see Supplementary Fig. 4 and Supplementary Note 1 for details).

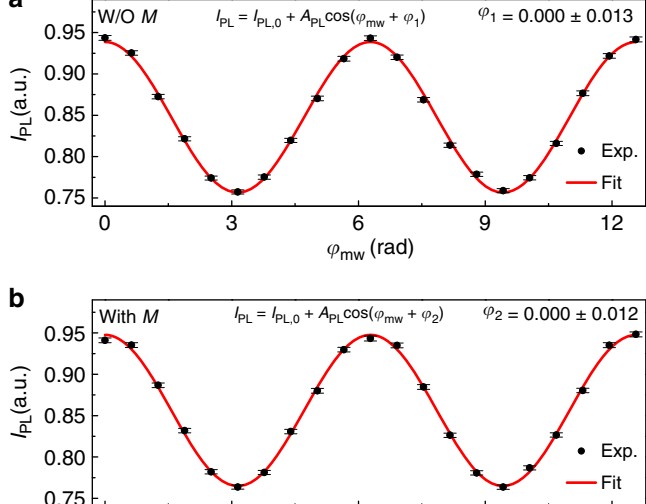

**Fig. 3** Experimental results for detecting the electron–nucleon interaction. **a** The measured photoluminescence intensity $I_{PL}$ without $M$. **b** The measured photoluminescence intensity $I_{PL}$ with $M$. In both panels, the experimental data are represented by black circles with error bars, and the red solid lines represent the fitting of the experimental data. Each experimental data is the average with six million experimental trails, which are divided into 1200 samples. Error bars of the experimental data represent s.e.m., which are calculated as the sample standard deviations divided by the square root of the sample size. The parameter values $A_{PL} = 0.091(1)$ and $I_{PL,0} = 0.8476$ (8) ($A_{PL} = 0.091(1)$ and $I_{PL,0} = 0.8563(8)$) are obtained by fitting the experimental data for the cases without $M$ in panel **a** (with $M$ in panel **b**). The phases $\varphi_1$ and $\varphi_2$ are the accumulated phases of the states of $S$ without and with $M$. The phase shift due to the electron–nucleon interaction between $S$ and $M$ is obtained by $\varphi = \varphi_2 - \varphi_1$ to be $\varphi = 0.000 \pm 0.018$ rad

Figure 2a shows schematically the distance $d$ and corresponding time-varying effective magnetic field $B_{eff}$ arising from the hypothetical electron–nucleon interaction. The mass is driven to vibrate with an angular frequency $\omega_m = 2\pi \times 187.29$ kHz. The vibration amplitude $A$ and shortest distance $d_0$ are $A = 41.1(1)$ nm and $d_0 = 0.5(1)$ μm, respectively. When $M$ vibrates to the position nearest to $S$, the distance $d$ reaches the minimum value $d_0$ and the corresponding effective magnetic field $B_{eff}$ achieves a maximum value. When $M$ vibrates to the position furthest from $S$, $d$ reaches the maximum value $d_0 + 2A$ and $B_{eff}$ achieves a minimum value.

Figure 2b shows the pulse sequence applied on $S$ (a detailed description of the pulse sequence is presented in Supplementary Fig. 3 and Supplementary Note 1) and the corresponding state evolution of $S$ on the Bloch sphere. The pulse duration of the $\pi$ ($\pi/2$) pulse is 118 ns (59 ns) and the waiting time $\tau$ is fixed to 2.67 μs. To optimize the phase accumulation, the microwave $\pi/2$ and $\pi$ pulses in the spin echo sequence are applied only when $M$ vibrates passing through the equilibrium point of the vibration. The electron spin $S$ is initialized into $|m_S = 0\rangle$ by a laser pulse, corresponding to the unit vector along $z$ axis in the Bloch sphere. The first microwave $\pi/2$ pulse transforms the state into $(|0\rangle - i|1\rangle)/\sqrt{2}$. Then $S$ evolves under the effective magnetic field $B_{eff}$ for half of the vibration period $\tau$, corresponding to the spin precessing around the $z$ axis. As a result, the state is evolved into $(|0\rangle - ie^{i\varphi_0}|1\rangle)/\sqrt{2}$ at the end of the free evolution, where $\varphi_0 = \int_{\tau/2}^{3\tau/2} \gamma B_{eff}(t) \cos\theta dt$ is the accumulated phase, and $\theta = \arccos(1/\sqrt{3})$ is the angle between $B_{eff}$ and the NV axis. The

following microwave $\pi$ pulse rotates the Bloch vector by an angle of $\pi$ around $x$ axis. After the $\pi$ pulse, the electron spin experiences another free evolution for half of the vibration period under $B_{eff}$. At the end of this evolution, the state is evolved into $(|0\rangle + ie^{-i\varphi}|1\rangle)/\sqrt{2}$ with $\varphi = \varphi_0 - \int_{3\tau/2}^{5\tau/2} \gamma B_{eff}(t) \cos\theta dt$. A final microwave $\pi/2$ pulse with phase $\varphi_{mw}$ then rotates the Bloch vector by an angle of $\pi/2$ around the axis $e_x \cos\varphi_{mw} + e_y \sin\varphi_{mw}$ (with $e_x$ and $e_y$ being the unit vector along the $x$ and $y$ axis), transforming the state into $\cos[(\varphi_{mw} + \varphi)/2]|0\rangle + e^{i\varphi_{mw}} \sin[(\varphi_{mw} + \varphi)/2]|1\rangle$. After this spin echo sequence, a laser pulse is applied and the photoluminescence intensity $I_{PL}$ is detected. The measured $I_{PL}$ reflects the population $P_{|0\rangle}$ of state $|m_S = 0\rangle$ for the final state, with $P_{|0\rangle} = 1/2 + 1/2 \cos(\varphi_{mw} + \varphi)$. Therefore, $I_{PL}$ can be expressed as

$$I_{PL} = I_{PL,0} + A_{PL} \cos(\varphi_{mw} + \varphi). \qquad (4)$$

By measuring the photoluminescence intensity $I_{PL}$ with a set of different phases $\varphi_{mw}$ of the final microwave $\pi/2$ pulse, we can extract $\varphi$ which contains the information of $B_{eff}$ arising from the spin–mass interaction. The coupling $g_s^N g_p^e$ can be derived to be

$$g_s^N g_p^e = \frac{1}{\cos\theta} \frac{2m}{\hbar\rho} \frac{\varphi}{\int_{\tau/2}^{3\tau/2} f(\lambda, R, d(t)) dt - \int_{3\tau/2}^{5\tau/2} f(\lambda, R, d(t)) dt}. \qquad (5)$$

**Experimental results**. Figure 3 shows the experimental results. All the experimental data shown in Fig. 3 are obtained with six million averages (see Supplementary Figs. 5, 6, and Supplementary Note 4 for details). To exclude the influence of any possible oscillating magnetic field from other sources, we first implement the pulse sequence without $M$ as a benchmark experiment. The experimental data without $M$ is shown in Fig. 3a. By fitting the data with Eq. (4), we obtain $\varphi_1 = 0.000 \pm 0.013$ rad as a benchmark. Then the spin echo sequence is implemented with vibrating $M$ and the result has been shown in Fig. 3b. The experimental data with $M$ is fitted with Eq. (4) to extract $\varphi_2$ with $\varphi_2 = 0.000 \pm 0.012$ rad. The accumulated phase $\varphi$ of the electron spin's state owing to $B_{eff}$ generated by $M$, which is obtained by $\varphi = \varphi_2 - \varphi_1$, is determined to be $\varphi = 0.000 \pm 0.018$ rad. The electron–nucleon interaction has not been observed at the current experimental condition, but an upper limit can be set to constrain the interaction.

Table 1 is the systematic error budget of our experiment. One systematic error is due to the diamagnetism of $M$ in a 300 G magnetic field. $M$ is modulated in phase with the spin echo sequence, so the in phase AC component rather than the DC component of magnetic field due to the diamagnetism of $M$ would cause a phase shift in our result. If the NV center locates exactly under the center of the mass, the magnetic field caused by the diamagnetism of $M$ is perpendicular to the NV symmetry axis, and the AC part of this magnetic field is estimated to be about $1.5 \times 10^{-6}$ G (see Supplementary Fig. 7 and Supplementary Note 4 for details). Due to the large energy splitting (2.0286 GHz) along the symmetry axis of NV center, the phase shift caused by this component is estimated to be $1.7 \times 10^{-10}$ rad. Because the NV center may deviate from the exact location under the center of the mass (see Supplementary Fig. 8 and Supplementary Note 4), there could be a residual magnetic field along the symmetry axis of NV. The amplitude of this in phase AC magnetic field is estimated to be about $1.1 \times 10^{-8}$ G (see Supplementary Note 4). Therefore, the correction to the $g_s^N g_p^e$ for 20 μm due to the diamagnetism of $M$ is $5(5) \times 10^{-20}$. The

**Table 1 Systematic error summary**

| Systematic error | Size of effect | Correction to $g_s^N g_p^e$ for 20 μm |
|---|---|---|
| Diamagnetism of $M$ | $-11.28 \times 10^{-6}$ | $(5 \pm 5) \times 10^{-20}$ |
| Diamagnetism of the tuning fork | $-11.28 \times 10^{-6}$ | $(3.8 \pm 0.3) \times 10^{-20}$ |
| Phase jitter of microwave | 1.3 ps | $(0.0 \pm 1.7) \times 10^{-27}$ |
| $T_2^*$ dephasing | $670 \pm 41$ ns | $(0.0 \pm 1.9) \times 10^{-27}$ |
| Shortest distance between $M$ and $S$ | $0.5 \pm 0.1$ μm | $(0.1 \pm 3.0) \times 10^{-17}$ |
| The amplitude of the modulation of $M$ | $41.1 \pm 0.1$ nm | $(0.0 \pm 1.3) \times 10^{-17}$ |
| The radius of $M$ | $250 \pm 2.5$ μm | $(0.1 \pm 3.7) \times 10^{-18}$ |
| The angle between $\mathbf{B}_{eff}$ and NV axis | $54.7 \pm 3^{\circ}$ | $(0.4 \pm 4.2) \times 10^{-16}$ |

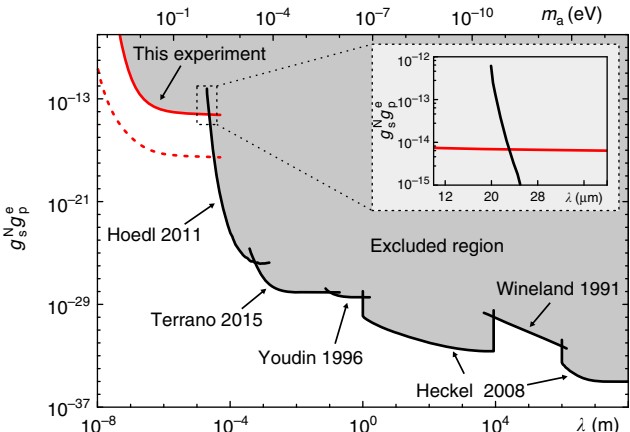

**Fig. 4** Upper limits on $g_s^N g_p^e$ as a function of the force range λ and mass of the axion-like particle $m_a$. Our result is represented as the red solid line. The black solid lines represent the results from refs. [8–12]. The red dashed line shows the available improvement of the constraint on $g_s^N g_p^e$ in future (see Supplementary Note 3 for details). The inset shows a comparison of our result and that from ref. [12] with the force range nearby 20 μm, which illustrates an improvement of two orders more stringent for our result at 20 μm compared with that from ref. [12]

material of the tuning fork is $SiO_2$. The distance between the tuning fork and the NV center is at least 250 μm. The systematic error due to the diamagnetism of tuning fork leads to a correction to $g_s^N g_p^e$ for 20 μm being $3.8(3) \times 10^{-20}$. The phase jitter of the microwave, which would cause the instability of the phase of the final $\pi/2$ pulse, is measured to be 1.3 ps (Supplementary Fig. 10 and Supplementary Note 4). Since the waiting time of the spin echo is fixed, this instability of the phase only causes a small reduction of the signal contrast rather than a phase shift. The impact of phase jitter is also presented in Table 1. The frequency shift of the microwave generator, the drift of the external magnetic field and the fluctuation of the Overhauser field (see Supplementary Fig. 9 and Supplementary Note 4) will contribute to the $T_2^*$ dephasing. This dephasing can be well suppressed by spin echo technique and the correction due to dephasing is also included in Table 1. The errors due to the uncertainties of the distance between $M$ and $S$, the amplitude of the modulation of $M$, the radius of $M$ and the angle between $\mathbf{B}_{eff}$ and NV axis, have also been taken into account in the Table 1. The detailed analysis of the systematic errors are included in Supplementary Note 4.

Figure 4 shows the new constraint set by this work together with recent constraints from experimental searches for monopole–dipole interactions[16]. The lines from the experiment by Heckel et al. are the upper limits in the meter range and above[10], except a gap from 10 to 1000 km. The upper limit in this gap is obtained by the experiment by Wineland et al.[8]. The experiment by Youdin et al. sets the upper limit in the range from

0.1 to 1 m[9]. The upper limit from the experiment by Terrano et al.[11] is for the range from 0.5 mm to 10 cm. In the range from 20 to 500 μm, the experiment by Hoedl et al.[12] provides the upper limit. Our result is represented as the solid red line. It is derived according to Eq. (5) with $2\delta_\varphi$ as an upper bound of $\varphi$, where $\delta_\varphi = 0.018$ rad is the s.d. of the accumulated phase $\varphi$. Besides $\delta_\varphi$, the uncertainties of other experimental parameters, such as $d_0$ and $A$, are also taken into account to derive the upper limit (see Supplementary Note 3 for details). For the force range 0.1 μm < λ < 23 μm, our result provided the upper bound for $g_s^N g_p^e$. As is shown in the inset of Fig. 4, the obtained upper bound of the interaction at 20 μm, $g_s^N g_p^e < 6.24 \times 10^{-15}$, is two orders of magnitude more stringent than the bound set by Hoedl et al.[12]. The possible value of mass of the ALPs, from $10^{-5}$ to 1 eV (corresponding to a force range 0.2 μm < λ < 2 cm), is still allowed by otherwise stringent constraints[23]. The unexplored force range left by the previous experiments has now been searched in our experiment. We note that the most restrictive constraint on $g_s^N g_p^e$ may arise from the combination of the long-range force bound and the astrophysical limit[16,24]. These limits rely on the underlying gravitational theory, namely, a chameleon mechanism could invalidate the astrophysical limit, and therefore, it is necessary to experimentally constrain $g_s^N g_p^e$ in laboratories, where the gravitational effects are negligible[25].

## Discussion

The constraint can be further improved by several strategies in future. We search for spin–mass interaction by detecting the accumulated phase of a single electron spin's state owing to $B_{eff}$. One effective method is to enhance the coherence time of the electron spin, by synthesizing $^{12}C$-enriched diamond[26] or by applying multi-pulse dynamical decoupling sequences[27,28]. Once the coherence time is prolonged, the ability of detecting the accumulated phase can be enhanced. The frequency of our tuning fork at present stage is 187.29 kHz, which is suitable for a spin echo sequence. If the frequency of the tuning fork is enhanced in future, multi-pulse dynamical decoupling sequences can be applied to improve the performance. On the other hand, the accumulated phase is proportional to the number density of nucleons in the source. To use materials with high number density of nucleons as the source, such as $Bi_4Ge_3O_{12}$ (BGO), can also improve the constraint. To decrease the measurement uncertainty of the accumulated phase, one can improve the detection efficiency of the photoluminescence and increase the number of experiment trails. On the basis of above extensions of techniques, the available constraint, which is shown as the red dashed line in Fig. 4, could be about three orders of magnitude improved from the current result (see detailed discussion in Supplementary Note 3).

Our platform uses a near-surface NV center together with AFM setup, thus the force range can be focused within micrometers. The micrometer and submicrometer range, which is not easily accessed in previous experiments, provides a new window for investigating new physics beyond standard model. The

electron–nucleon interaction investigated in our work is one of interactions from new particle exchange[17]. In future, several related interactions can also be investigated with extension of our method. For example, spin–spin interaction mediated by ALPs, on which a constraint is recently set at micrometer scale[29], can be further explored with submicrometer scale by two coupled NV centers with technologies developed by Grinolds et al.[30]. Another case is to explore the interaction mediated by a vector boson, which has been investigated at micrometer force range[31,32]. Therefore, NV centers will not only be a promising quantum sensor for physics within standard model[33–38], but also be an important platform for searching for new particles predicted by theories beyond the standard model.

## Methods

**Experimental setup.** The electron spin of a near-surface NV center in diamond is used as a quantum sensor to search for the hypothetical ALP-mediated monopole–dipole interaction with nucleons in a half-ball lens. The NV center was created by implantation of 10 keV $N_2^+$ ions into [100] bulk diamond and annealing for 2 h at 800 °C in vacuum. The diamond was then oxidatively etched for 4 h at 580 °C. The depth of the NV center was estimated to be <10 nm. Nanopillars were fabricated to improve the detection efficiency of the photoluminescence, with which a photoluminescence rate of 100 kcounts s$^{-1}$ was achieved in the experiment. The NV center is confirmed to be single by measurement of the second-order correlation function (Supplementary Fig. 2 and Supplementary Note 1). An optically detected magnetic resonance setup combined with an AFM, which is similar with setup reported in ref.[39], was constructed to search for the spin–mass interaction. The 532 nm green laser pulse passed through an acousto-optic modulator and an objective to be focused on the NV center to initialize the electron-spin state. The phonon sideband fluorescence with wavelength of 650–800 nm went through the same objective and was collected by an avalanche photodiode with a counter card to realize state readout. Microwave pulses, which were generated by IQ modulation with a 4.2 GSa s$^{-1}$ arbitrary waveform generator (Keysight 81180A) and a vector signal generator (Keysight E8267D) were amplified by a power amplifier (Mini-Circuits ZHL-16W-43-S+) and delivered by a copper microwave wire to manipulate the electron-spin state. The tuning fork-based atomic force microscope was utilized to position the half-ball lens and to drive the half-ball lens to vibrate. The state initialization, manipulation, and readout of the electron spin were synchronized with the vibration of the half-ball lens with an arbitrary sequence generator (Hefei Quantum Precision Device Co. ASG-GT50-C). Details of the experimental setup are shown in Supplementary Fig. 1 and Supplementary Note 1.

**Data availability.** Data supporting the findings of this study are available within the article and its Supplementary Information file, and from the corresponding authors upon reasonable request.

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

## Acknowledgements

We are grateful to H.Y. Yan for his systematic introduction about the spin-dependent forces and fruitful discussion about the experiment. We thank C.K. Duan and D.J. Kimball for helpful discussion. We thank L.P. Guo for his help on nitrogen ion implantation. The fabrication of diamond nanopillars for improving the detection efficiency of the photoluminescence was performed at the USTC Center for Micro and Nanoscale Research and Fabrication. This work was supported by the National Key Basic Research Program of China (Grants Nos. 2013CB921800, 2016YFA0502400, and 2016YFB0501603), the National Natural Science Foundation of China (Grant Nos. 11227901, 91636217, 11722327, and 31470835) and the Strategic Priority Research Program (B) of the CAS (Grant No. XDB01030400). J.D. and X.R. thank financial support by Key Research Program of Frontier Sciences, CAS (Grants No. QYZDY-SSW-

SLH004 and QYZDB-SSW-SLH005). F.S. and X.R. thank the Youth Innovation Promotion Association of Chinese Academy of Sciences for the support. Y.-F.C. is supported in part by the Chinese National Youth Thousand Talents Program, by the CAST Young Elite Scientists Sponsorship Program (2016QNRC001), by the National Natural Science Foundation of China (Grant Nos. 11421303, 11653002), and by the Fundamental Research Funds for the Central Universities. X.Q. thank support by Fundamental Research Funds for the Central Universities (Grant No. WK2030040081).

## Author contributions

J.D. proposed the idea. J.D. and X.R. designed the experiment. M.W., J.G., and M.G. performed the experiment under the supervision of J.D. and X.R., J.G., M.W., M.J., and P.H. carried out the calculations and simulations. X.Q., P.W., M.G., Y.X., and F.S. constructed the experimental setup. M.W. and C.Z. prepared the NV center. X.R., J.G., M.W., Y.-F.C., and M.G. wrote the paper. All authors analyzed the data, discussed the results and commented on the manuscript.

## Additional information

**Competing interests:** The authors declare no competing financial interests.

