## [Peer Review File · Nature Communications]

Reviewers' comments:

Reviewer #1 (Remarks to the Author):

Report on „Searching for new particles beyond the standard model with a single electron-spin quantum sensor“

The method shown in this paper to search for scalar pseudo-scalar electron spin interaction with exotic particles is really compelling. It could open the direct searchable window to larger masses of the interacting bosons, in most cases ALPs (axion like particle). Nevertheless the paper has a big flaw. The discussion of systematic and statistical error/uncertainties and a detailed error budget is missing, which is not acceptable. Without discussion of all systematic errors the community cannot judge on the reliability of the result and its validity in spite of possible false effects, which may be recognized after publication.

Therefore I can only recommend a major revision of this paper before the decision of publication, i.e. adding a detailed discussion of systematic and statistical errors.

More in detail:

Just from a few back of the envelope calculations the following questions arise. Any change of the order of 10nT of the magnetic field at the NV center would cause a comparable phase shift as you want to detect respectively exclude. Of course, only the Fourier component which is in phase with your spin echo sequence will contribute to your signal. But the moving mass M is total in phase with your spin echo sequence because the spin echo sequence is synchronized to the mass movement. The mass M moves in a magnetic field of 30mT according to your setup description in concordance with the driving frequency in the GHz range of the spin echo pulses. Even diamagnetism of the material of the moving mass M with a susceptibility 10^{-6} can cause such magnetic field changes. So, how can you exclude magnetic effects caused by the moving mass including the complete driving stage?

At page 4 first section, you talk about dephasing effects limiting the Ramsey method. You mentioned in this context static magnetic field drifts. With the spin echo method magnetic field drifts are only partly canceled. More precisely, only the even parts of a Taylor expansion around the Pi-flip point are canceled, the odd ones not. Therefore phase shifts caused by static magnetic field drifts have to be analyzed and taken into account for the systematic error budget.

How are the errors/uncertainties shown in the data figure 3 are calculated? Does they include systematic errors or do they represent only the counting statistics? I strongly recommend showing these plots with real measured units, which are counts according to figure S1, in place of arbitrary units. At least the contrast of your measurement $A(PLI)$ to $I(PL,0)$ should be mentioned. Also, for people who are familiar with spin echo techniques, it would be nice to have on the upper horizontal frame the delay time, corresponding to phase shift plotted on the lower one.

How are the NV centers distributed in the diamond plate? I guess from the production method, ion implantation, that there more than one NV center in the plate. How fare they are apart from each other? Is it sure, that only one NV center is within the focus of the green laser light? How well could the alignment of the NV center with respect to the moving mass be done?

Why there are only the results of one measurement run and one calibration run of phase zero run are shown? Normally the result of at least 5 runs has to be shown in order to demonstrate the reproducibility and to get a rough estimate on the variability of the result. How long does one measurement run take? To my opinion, one measurement run is enough to demonstrate the feasibility of the method, but it is by far not sufficient to base a physics result on it.

Finally there is an important newer result missing in the exclusion plot figure 4: See Phys. Rev. Lett 115 (2015) 201801. Reference 16 of your paper (from 2012) is no longer up to date. Further it should be mentioned that laboratory searches for non newtonian gravity and equivalence principle tests, i.e. measuring $g(\text{Nucleon})$ scalar coupling in combination with stellar models of red giant star evolution, i.e. an astrophysical limit for $g(\text{electron})$ pseudo scalar coupling, leads to much lower limits for the combined coupling $g(\text{Nucleon})$ scalar times $g(\text{electron})$ pseudo scalar, as shown in reference 16 of your paper.

Minor questions:

Please quantify the phase jitter and frequency shifts of Micro wave generation over the duration of a whole data taking run. What is their impact on the final phase measurement?

Reviewer #2 (Remarks to the Author):

Rong et al describe an experiment that searches for a new CP-violating electron spin-nucleon coupling, as could be generated by the virtual exchange of an undiscovered pseudoscalar particle. The manuscript claims to set the tightest upper limit on such a force over the length scale of 0.1-23 microns. The experimental approach is novel in the field of short distance tests of new spin coupled forces in that it employs a diamond NV center located with nm's of the diamond surface. I believe the technique described will be of interest to others in the new force and magnetometry communities.

The manuscript is well written and the motivation for the work is clearly presented. A final pass for typos should be made (eg lase pulse in the first paragraph on p.5 should be laser pulse). Apart from the comments that follow, I find Rong et al to be a nice piece of work that deserves to be published.

My major concern with the manuscript is an insufficient discussion of systematic errors. Systematic errors usually limit the sensitivity of weak force measurements. A back of the envelope estimate suggests to me that the diamagnetic susceptibility of the half sphere should perturb the ambient 300 Gauss magnetic field at the level of 3 milli-Gauss, with a $1/r^3$ distance dependence, producing a signal that is of order of the experimental sensitivity. The vibrating AFM tip could make this worse, although no details about the composition of the tuning fork were give. Yet the experimental results show exactly zero phase shift for both measurements, with and without the silica half sphere. A phase shift of even 1 mrad would require the length of periods before and after the pi-pulse to be the same within 0.2 pico-seconds and high field stability (because of the high rate of phase accumulation in the 300 Gauss background field). I believe the authors should add a discussion about potential systematic errors due to magnetic susceptibilities and say more about the timing sequence parameters that allowed them to unwind the magnetic phase accumulation so effectively.

In summary, I am impressed by the experiment described in Rong et al. The results are likely to be correct, but warning flags are raised when the measurement results are "too clean". For me to recommend publication, I need to be convinced that magnetic susceptibility is not a problem and how sub pico-second timing was achieved. A suggestion would be to add the terms from the 300 Gauss field to the phase equations, and show how the cancellation between the 2 periods sets requirements on field stability and pulse timing. (I am not an expert on the rf techniques that were employed, but neither will be most of the readers of the manuscript.)

Reviewer #3 (Remarks to the Author):

The authors Xing Rong et al. report on an experiment for the search of fundamental particles using the electron spin associated with a single nitrogen vacancy center near the surface of a bulk

diamond crystal. The article is well written and the experiment is carefully described. As the main result, a new upper bound for electron-nuclear coupling with a new force range in the μm regime is obtained. I have the following questions and comments to the article.

- 1) The applied pulse sequence used for the detection of an oscillating magnetic field is described in detail. For the case of nitrogen-vacancy (NV) centers in diamond, this principle was first suggested and proposed by Taylor et al. in Nat. Phys. 4, 810–816 (2008). The reference should be provided in the article.
- 2) Conceptually, a similar experiment has been carried out with a magnetic atomic force microscope cantilever and a NV center, 1. Kolkowitz, S. et al., Science 335, 1603-1606 (2012), but the reference is also not cited in the present version.
- 3) The application of a single re-focusing π pulse is a standard technique in all laboratories handling NV centers and quantum physical. Hence, in my opinion a lengthy description as provided here is not required. However, essential physical parameters, such as Rabi-frequency (characterizing how fast the spin is swapped) and readout-contrast, characterizing the setup and the physical performance should be provided instead.
- 4) It is mentioned in the discussion section and the supplementary material, that higher order pulse sequences such as CPMG may be used to even further improve the constraint. Since the authors are equipped with the necessary electronics, in particular a fast AWG, I do not understand why this has not been done in the present work. Many NV center related articles report on the application of higher order sequences for field detection in the kHz-MHz frequency range. If the application of higher order sequences is for some reasons not possible with the present NV center system, for instance due to the relatively low eigen-frequency ω_m of the tuning fork in combination with the NV coherence, then a more critical discussion would be appreciated and should be presented in the main article.
- 5) I would think that the phase between the tuning fork oscillation and the pulse sequence should be very well matched. How did the authors ensure that indeed their sequence is very well phase matched to the oscillation of the tuning fork?
- 6) The tuning fork itself is also made of a mass oscillating with ω_m , but at a larger distance though. What is the impact of this with respect to the context of the present experiment?

In conclusion, while I'm convinced that the present article certainly warrants publication, I do not think it of great interest to the broad readership of Nature Communications. Rather I think the article is relevant for a more specialized journal. Furthermore, the measurement sequence can be improved by the application of a higher order sequence. This was not done for reasons, which are not obvious from the article.

Smaller comments are:

- On page 5, the noise arising from fluctuating nuclear spins is referred to as Overhauser field, instead of Overhouser field.
- On page 5, dephasing is typically an unwanted observation, but not an effect.
- In my opinion, it is more convenient to quote angular frequencies in units of $[2\pi \text{ Hz}]$, which is easier to relate to the coherence times.
- Fig. 3 plots the intensity vs the phase of the readout pulse. Although it's a re-normalized signal, but an intensity should not be negative. Rather the state population of either $m_s=-1$ or $m_s=0$ should be plotted.
- In Fig. 5, instead of numbering the traces from 1 to 5, I would think it is more convenient to insert the references directly.

The point-by-point response to the referees' comments is shown in the following (the referees' comments in roman font and our replies in italic).

Reviewer #1 (Remarks to the Author):

Report on "Searching for new particles beyond the standard model with a single electron-spin quantum sensor"

The method shown in this paper to search for scalar pseudo-scalar electron spin interaction with exotic particles is really compelling. It could open the direct searchable window to larger masses of the interacting bosons, in most cases ALPs (axion like particle). Nevertheless the paper has a big flaw. The discussion of systematic and statistical error/uncertainties and a detailed error budget is missing, which is not acceptable. Without discussion of all systematic errors the community cannot judge on the reliability of the result and its validity in spite of possible false effects, which may be recognized after publication.

Therefore I can only recommend a major revision of this paper before the decision of publication, i.e. adding a detailed discussion of systematic and statistical errors.

We thank the referee for the suggestion on a detailed discussion of systematic and statistical errors. In the revised version, a detailed systematic error budget and discussion have been presented in the revised main text and supplementary information (SI).

Just from a few back of the envelope calculations the following questions arise. Any change of the order of 10nT of the magnetic field at the NV center would cause a comparable phase shift as you want to detect respectively exclude. Of course, only the Fourier component which is in phase with your spin echo sequence will contribute to you signal. But the moving mass M is total in phase with your spin echo sequence because the spin echo sequence is synchronized to the mass movement. The mass M moves in a magnetic field of 30mT according to your setup description in concordance with the driving frequency in the GHz range of the spin echo pulses. Iven diamagnetism of the material of the moving mass M with a susceptibility 10^{-6} can cause such magnetic field changes. So, how can you exclude magnetic effects caused by the moving mass including the complete driving stage?

We appreciate referee for reminding us to consider the effect of the susceptibility of the mass. Actually, this effect is negligible according to the following analyses.

1) We utilized the spin echo technique to acquire the in phase modulated magnetic field signal. Our analysis shows that the main component of the magnetic field caused by the diamagnetism of M is perpendicular to the NV symmetry axis. The d.c. part of this magnet field is about 1.4 mG, which introduces a very small energy shift (about

0.008 Hz) along the symmetry axis of NV center due to the large energy splitting ~ 2.0286 GHz. The influence of this d.c. component is cancelled by applying the spin echo sequence. The a.c. component of this magnetic field is about $1.5E-6$ G, corresponding to a $1.6E-5$ Hz energy shift along the NV symmetry axis. And the phase shift caused by this a.c. component in our experiment is estimated to be $1.7 E-10$ rad. This can be negligible.

2) The NV center may deviate from the exact location under the center of the bottom of M. The deviation is measured to be $0.7(8) \mu\text{m}$ (see Section 4 in the revised SI). Therefore, apart from the magnetic field considered above, there could be a residual magnetic field along the NV symmetry axis. The amplitude of the in phase a.c. magnetic field along the symmetry axis of NV is estimated to be $1.1E-8$ G, which is much lower than the sensitivity of our quantum sensor.

Consequently, the correction to the bound at $20 \mu\text{m}$ due to diamagnetism of M is estimated to be $5E-20$, which is much below our experimental bound. In the revised version, we have followed referee's suggestion to include this effect in a detailed error budget (Table 1 in the revised manuscript and see list of changes, 1). The detailed calculation and analysis are included in the SI (see list of changes, 6).

At page 4 first section, you talk about dephasing effects limiting the Ramsey method. You mentioned in this context static magnetic field drifts. With the spin echo method magnetic field drifts are only partly canceled. More precisely, only the even parts of a Taylor expansion around the Pi-flip point are canceled, the odd ones not. Therefore phase shifts caused by static magnetic field drifts have to be analyzed and taken into account for the systematic error budget.

We thank referee for this suggestion. We analyzed the effect of the magnetic field drift and took it into account for the systematic error budget in the revised version. The slow drift of the magnet field can be taken as the quasi-static fluctuation of the magnetic field. Its effect is cancelled by applying the spin echo technique [Nat. Commun. 3, 997 (2012), PRL 112, 050503 (2014)]. Note that the waiting time, τ , in the spin echo sequence is fixed, and therefore, the fast drift of magnetic field contributes to the reduction of the signal contrast, rather than a phase shift [PRL 107, 230501 (2011)]. The effect due to the drift magnetic field has also been included in the error budget (Table 1 in the revised main text).

How are the errors/uncertainties shown in the data figure 3 are calculated?

The uncertainties in the data points are standard deviations.

Does they include systematic errors or do they represent only the counting statistics?

The error in the experimental data is due to the counting statistics. The uncertainties in the extracted phases (φ_1 and φ_2) are obtained from the fitting errors. The systematic errors in the budget (Table 1 in the revised main text) are much smaller than the fitting errors.

I strongly recommend showing these plots with real measured units, which are counts according to figure S1, in place of arbitrary units. At least the contrast of you measurement A(PLI) to I(PL,0) should be mentioned.

In the revised main text, Figure 3 is re-plotted with real measured data rather than rescaled data according to referee's suggestion (see list of changes, 2). The values of A(PLI) and I(PL,0) in our measurement have been mentioned in the revised version.

Also, for people who are familiar with spin echo techniques, it would be nice to have on the upper horizontal frame the delay time, corresponding to phase shift plotted on the lower one.

We realized that our previous description of the experiment might have caused some misunderstanding. Actually the waiting time in the spin echo pulse sequence is fixed, and the phase plotted on the lower horizontal frame is the microwave phase of the last $\pi/2$ pulse, which is not related to the waiting time. This type of phase measurement is similar with the Ramsey interferometry [PRB 90, 024422 (2014), Science Advances 2, e1501732 (2016) and Nature Communications 9, 697(2017)]. The detailed information about the experimental spin echo sequence has been provided in the revised version (see list of changes, 3).

How are the NV centers distributed in the diamond plate? I guess from the production method, ion implantation, that there more than one NV center in the plate. How fare they are apart from each other? Is it sure, that only one NV center is within the focus of the green laser light?

The average separation between NV centers is of about $2 \mu\text{m}$. The figure of the distribution of the NV centers is shown in Figure S2 in the revised SI (see list of changes, 6). To ensure that only one NV center is within the focus of the green laser light, the second-order correlation function $g^{(2)}(\tau)$ has been measured. The result gives $g^{(2)}(0)=0.13$, confirming that it is a single NV center [Opt. Lett. 25, 1294(2000)].

How well could the alignment of the NV center with respect to the moving mass be done?

The NV center may deviate from the exact location under the center of the bottom of M. The deviation is measured to be $0.7(8) \mu\text{m}$ (see Section 4 in the revised SI). This leads to a residual magnetic field along the symmetry axis of NV, whose effect on the measured phase is negligible as discussed above.

Why there are only the results of one measurement run and one calibration run of phase zero run are shown? Normally the result of at least 5 runs has to be shown in order to demonstrate the reproducibility and to get a rough estimate on the variability of the result. How long does one measurement run take? To my opinion, one measurement run is enough to demonstrate the feasibility of the method, but it is by far not sufficient to base a physics result on it.

The whole experiment contains 6 separate runs, with each run contains one million trails. Each trail takes about 20 microseconds. We present all runs separately in the revised SI (Figure S5 and S6). In the main text, we present the final result taking all the six runs into consideration. The reproducibility of our experiment is affirmed from six separated runs.

Finally there is an important newer result missing in the exclusion plot figure 4: See Phys. Rev. Lett 115 (2015) 201801. Reference 16 of your paper (from 2012) is no longer up to date. Further it should be mentioned that laboratory searches for non newtonian gravity and equivalence principle tests, i.e. measuring $g(\text{Nucleon})$ scalar coupling in combination with stellar models of red giant star evolution, i.e. an astrophysical limit for $g(\text{electron})$ pseudo scalar coupling, leads to much lower limits for the combined coupling $g(\text{Nucleon})$ scalar times $g(\text{electron})$ pseudo scalar, as shown in reference 16 of your paper.

We thank referee for this suggestion. The reference has been updated, and the Figure 4 and the related discussion (the last paragraph in page 8 of revised main text) have been revised according to this suggestion (see list of changes 2 and 4).

Minor questions:

Please quantify the phase jitter and frequency shifts of Micro wave generation over the duration of a whole data taking run. What is their impact on the final phase measurement?

The impact of phase jitter and frequency shifts of microwave is negligible due to the following reasons.

1) For the phase jitter, it will cause the instability of the phase of the final $\pi/2$ pulse. The measured phase jitter is 1.3 ps. Since the time of the spin echo sequence is fixed,

this phase jitter mainly deteriorate the signal contrast. The correction to spin-mass interaction for 20 micrometers due to phase jitter is $(0.0 \pm 1.7) \times 10^{-27}$, which is negligible. In the revised version, we present the detailed analysis.

2) The frequency shifts will introduce an additional term $\delta\sigma_z$ in the spin Hamiltonian. The slow drift can be cancelled by the spin echo technology. The fast frequency drift only leads to deterioration of the signal contrast, because the waiting time of the spin echo sequence is fixed. We discussed this in the revised SI.

In the revised version, we analyze their impact on the final phase measurement in the SI. Their impact on our result is also included in Table 1 in the revised version (see list of changes, 1 and 6).

Reviewer #2 (Remarks to the Author):

Rong et al describe an experiment that searches for a new CP-violating electron spin-nucleon coupling, as could be generated by the virtual exchange of an undiscovered pseudoscalar particle. The manuscript claims to set the tightest upper limit on such a force over the length scale of 0.1-23 microns. The experimental approach is novel in the field of short distance tests of new spin coupled forces in that it employs a diamond NV center located with nm's of the diamond surface. I believe the technique described will be of interest to others in the new force and magnetometry communities.

The manuscript is well written and the motivation for the work is clearly presented. A final pass for typos should be made (eg lase pulse in the first paragraph on p.5 should be laser pulse). Apart from the comments that follow, I find Rong et al to be a nice piece of work that deserves to be published.

We thank referee for her/his recommendation. The typos have been corrected in the revised version.

My major concern with the manuscript is an insufficient discussion of systematic errors. Systematic errors usually limit the sensitivity of weak force measurements. A back of the envelope estimate suggests to me that the diamagnetic susceptibility of the half sphere should perturb the ambient 300 Gauss magnetic field at the level of 3 milli-Gauss, with a $1/r^3$ distance dependence, producing a signal that is of order of the experimental sensitivity. The vibrating AFM tip could make this worse, although no details about the composition of the tuning fork were give. Yet the experimental results show exactly zero phase shift for both measurements, with and without the silica half sphere.

We add a detailed discussion of systematic errors in the revised version. The diamagnetic effect of the half sphere and the tuning fork have been taken into account. All these contributions can be neglected according to the following analyses.

1) We utilized the spin echo technique to acquire the in phase modulated magnetic field signal. Our analysis shows that the main component of the magnetic field caused by the diamagnetism of M is perpendicular to the NV symmetry axis. The d.c. part of this magnet field is about 1.4 mG, which introduces a very small energy shift (about 0.008 Hz) along the symmetry axis of NV center due to the large energy splitting ~ 2.0286 GHz. The influence of this d.c. component can be cancelled by the spin echo sequence. The a.c. component of this magnetic field is about 1.5×10^{-6} G, corresponding to a 1.6×10^{-5} Hz energy shift along the NV symmetry axis. And the phase shift caused by this a.c. component is estimated to be 1.7×10^{-10} rad. This can be negligible.

2) The NV center may deviate from the exact location under the center of the mass. The deviation is measured to be $0.7(8) \mu\text{m}$ (see Section 4 in the revised SI). Therefore, apart from the magnetic field considered above, there could be a residual magnetic field along the NV symmetry axis. The amplitude of the in phase a.c. magnetic field along the symmetry axis of NV is estimated to be 1.1×10^{-8} G, which is much lower than the sensitivity of our quantum sensor.

3) The material of the tuning fork is SiO_2 . The distance between the NV center and the tuning fork is at least $250 \mu\text{m}$. The diamagnetic effect of the tuning fork is also negligible. The in phase a.c. magnetic field along NV axis due to the tuning fork is about 8.5×10^{-9} Gauss, which is far below the sensitivity of our setup.

In summary, the correction to the spin-mass interaction for $20 \mu\text{m}$ due to the diamagnetic effect of the half sphere (the tuning fork) is 5×10^{-20} (3.8×10^{-20}). The detailed discussion is shown in the revised version (see list of changes 1 and 6).

A phase shift of even 1 mrad would require the length of periods before and after the pi-pulse to be the same within 0.2 pico-seconds and high field stability (because of the high rate of phase accumulation in the 300 Gauss background field).

The phase accumulation in the laboratory frame would require a precision of time within 0.2 pico-seconds. Fortunately, in our experiment, like most of NMR and ESR experiments, the phase accumulation in the rotating frame rather than that in the laboratory frame is measured. The rate of phase accumulation in the rotating frame depends on the microwave frequency detuning. Even a large detuning of 0.1 MHz, requires 1.6 ns to accumulate 1 mrad. The time precision of our experimental setup is 0.24 ns and so the time resolution of electronic is sufficient. This point has been discussed in detail in the revised SI (see list of changes 6).

I believe the authors should add a discussion about potential systematic errors due to magnetic susceptibilities and say more about the timing sequence parameters that allowed them to unwind the magnetic phase accumulation so effectively.

In the revised version, we presented a detailed error budget and detailed discussion about these (see list of changes, 1 and 6). The effect of magnetic susceptibilities and the timing sequence parameters are also discussed in detail in the revised version.

In summary, I am impressed by the experiment described in Rong et al. The results are likely to be correct, but warning flags are raised when the measurement results are "too clean". For me to recommend publication, I need to be convinced that magnetic susceptibility is not a problem and how sub pico-second timing was achieved. A suggestion would be to add the terms from the 300 Gauss field to the phase equations, and show how the cancellation between the 2 periods sets requirements on field stability and pulse timing. (I am not an expert on the rf techniques that were employed, but neither will be most of the readers of the manuscript.)

We are grateful of the referee's recommendation and constructive suggestions. The effect due to the diamagnetic susceptibility of the half sphere and the tuning fork under 300 Gauss is negligible as discussed above. According to the detailed analysis in section 1 of the revised SI, the sub pico-second timing is not necessary in our experiment.

Reviewer #3 (Remarks to the Author):

The authors Xing Rong et al. report on an experiment for the search of fundamental particles using the electron spin associated with a single nitrogen vacancy center near the surface of a bulk diamond crystal. The article is well written and the experiment is carefully described. As the main result, a new upper bound for electron-nuclear coupling with a new force range in the μm regime is obtained. I have the following questions and comments to the article.

1) The applied pulse sequence used for the detection of an oscillating magnetic field is described in detail. For the case of nitrogen-vacancy (NV) centers in diamond, this principle was first suggested and proposed by Taylor et al. in Nat. Phys. 4, 810–816 (2008). The reference should be provided in the article.

2) Conceptually, a similar experiment has been carried out with a magnetic atomic force microscope cantilever and a NV center, 1. Kolkowitz, S. et al., Science 335, 1603-1606 (2012), but the reference is also not cited in the present version.

These references have been cited in the revised version (see list of changes, 5).

3) The application of a single re-focusing π pulse is a standard technique in all laboratories handling NV centers and quantum physical. Hence, in my opinion a lengthy description as provided here is not required. However, essential physical parameters, such as Rabi-frequency (characterizing how fast the spin is swapped) and readout-contrast, characterizing the setup and the physical performance should be provided instead.

In the revised version, we have refined the description of the spin echo sequence and provided the parameters about the experiment (see list of changes, 3).

4) It is mentioned in the discussion section and the supplementary material, that higher order pulse sequences such as CPMG may be used to even further improve the constraint. Since the authors are equipped with the necessary electronics, in particular a fast AWG, I do not understand why this has not been done in the present work. Many NV center related articles report on the application of higher order sequences for field detection in the kHz-MHz frequency range. If the application of higher order sequences is for some reasons not possible with the present NV center system, for instance due to the relatively low eigen-frequency ω_m of the tuning fork in combination with the NV coherence, then a more critical discussion would be appreciated and should be presented in the main article.

The main limitation of our present setup is the low frequency of the tuning fork, which has now been discussed in the first paragraph in Discussion section of the revised main text. However, this issue can be resolved with some efforts in future. Once the frequency of the tuning fork is enhanced, high order CPMG can be applied. We have revised the manuscript to clarify this point (see list of changes, 3).

5) I would think that the phase between the tuning fork oscillation and the pulse sequence should be very well matched. How did the authors ensure that indeed their sequence is very well phase matched to the oscillation of the tuning fork?

The pulse sequence and the oscillation of tuning fork are synchronized by an arbitrary sequence generator (ASG) and a comparator, which has been discussed in Section 1 of SI. The vibration of tuning fork will generate an oscillating electronic signal due to the piezoelectric effect. This oscillating signal is transformed into a periodic rectangular pulse train by a comparator. The rectangular pulse train triggers the ASG, which generates rectangular pulses to trigger the arbitrary waveform generator.

In the revised version of SI, we present the experimental pulse sequence together with the oscillating signal from the tuning fork (Figure S4). The experimental data show

that the sequence has been well synchronized to the oscillation of the tuning fork (see list of changes, 3 and 6).

6) The tuning fork itself is also made of a mass oscillating with ω_m , but at a larger distance though. What is the impact of this with respect to the context of the present experiment?

The tuning fork could introduce systematic errors, but this effect is negligible based on the following reasoning. The phase shift can be caused by the a.c. component of magnetic field due to diamagnetism of the tuning fork. The material of the tuning fork is SiO₂. Because of the large distance (>250 μm) and very small vibration amplitude (41.1 nm), the amplitude of this a.c. component of magnetic field along the NV symmetry axis is about 8.5E-9 and the corresponding phase shift is negligible. The detailed discussion on the impact of the tuning fork is shown in the revised version (see list of changes 1 and 6).

In conclusion, while I'm convinced that the present article certainly warrants publication, I do not think it of great interest to the broad readership of Nature Communications. Rather I think the article is relevant for a more specialized journal. Furthermore, the measurement sequence can be improved by the application of a higher order sequence. This was not done for reasons, which are not obvious from the article.

While we appreciate a lot for the positive comments given by Referee 3, we would like to address the concerns on the broadness of the current study. Searching for the axions and axionlike particles by utilizing the technique of quantum measurements is of sufficient interest to a very broad readership of Nature Communications. Firstly, these types of bosonic scalars are considered as key elements to face at the deficiencies of the current paradigm of Particle Physics, Astrophysics and Cosmology, namely, the hierarchy problem, the strong CP problem, matter-antimatter asymmetry, and possible connections to dark matter and dark energy. Secondly, the technique of using a microscopic quantum sensor to search for them is crucial for exploring new physics beyond the Standard Model of particle physics, which has become extremely challenging using the traditional experiments based on particle colliders, such as the Large Hadron Collider. Our work belongs to those which probe the frontiers of new particles and forces with tabletop-scale experiments, which have been discussed very recently in Science [Science 357, 990 (2017)]. Thirdly, with our attempt, a major development has been achieved for limiting the spin-mass interactions by using the quantum sensing based on NV center, which is novel both in quantum sensing by NV center and in the field of searching for new spin-dependent interactions.

Just as the other two referees' comments, our work "could open the direct searchable window to larger masses of the interacting bosons, in most cases ALPs (axion like particle)"; "the experimental approach is novel in the field of short distance tests of

new spin coupled forces” and “the technique described will be of interest to others in the new force and magnetometry communities”. Therefore, from aforementioned arguments, we believe that our current work meet the high-level criterions of Nature Communications in the aspects of broad readership, general interest, as well as significance.

Smaller comments are:

- On page 5, the noise arising from fluctuating nuclear spins is referred to as Overhauser field, instead of Overhouser field.
- On page 5, dephasing is typically an unwanted observation, but not an effect.
- In my opinion, it is more convenient to quote angular frequencies in units of $[2\pi \text{ Hz}]$, which is easier to relate to the coherence times.

We thank referee for the comments. The manuscript has been revised accordingly (see list of changes, 5).

- Fig. 3 plots the intensity vs the phase of the readout pulse. Although it's a re-normalized signal, but an intensity should not be negative. Rather the state population of either $m_s=-1$ or $m_s=0$ should be plotted.

We thank referee for the reminder. In the revised version, Figure 3 is re-plotted following referee 1's suggestion (see list of changes, 2).

- In Fig. 5, instead of numbering the traces from 1 to 5, I would think it is more convenient to insert the references directly.

Figure 4, which includes the constraints, has been revised according to this suggestion, and the discussion in the main text has been modified accordingly (see list of changes, 2 and 4).

List of changes

- 1. Page7: A table, which presents the systematic errors, has been added in the main text. A paragraph, which discusses the systematic error budget, has also been added.*
- 2. Figure 3 and 4 in the main text have been revised according to referees' suggestions. The related discussions in the main text have been updated.*

3. We add “We use a homebuilt pulse generator and a comparator to make sure that the tuning fork oscillation and the pulse sequence are synchronized well (see SI for details).” in the end of the second paragraph of page 5. We add “The pulse duration of the π ($\pi/2$) pulse is 118 ns (59 ns) and the waiting time τ is fixed to $2.67\mu\text{s}$ ” in the first paragraph of page 6. We add “The frequency of our tuning fork at present stage...to improve the performance” according to referee’s suggestion in the first paragraph of the discussion.

4. Page 8, last paragraph: we updated recent constraints from experimental searches for monopole-dipole interactions according to referee’s suggestion. We add “We note that the most restrictive constraint on ...where the gravitational effects are negligible” at the end of this paragraph.

5. References have been added in the proper positions in the revised main text according to referees’ suggestions. Various other minor revisions to correct typos, to improve readability etc.

6. Revisions to the SI to clarify various related issues:

1) Section 1: We add the confocal image of the NV center (Figure S2 a). The second-order correlation function measurement of the NV center in our experiment has been presented (Figure S2 b). The detailed experimental pulse sequence is shown in Figure S3. The experimental time sequence in Figure S4 shows that that the trigger of laser pulses, the microwave pulse sequences and the moving of the tuning fork are synchronized very well. The Hamiltonian of NV center system under 300 Gauss has been presented. The requirement for the time resolution of our electronic system have been discussed.

2) We add Section 4, which includes six separate runs of experiment with and without M (in Figure S5). Figure S6 includes the obtained phase differences from these experiments. The systematic and statistical errors have been discussed in detail.

Reviewers' comments:

Reviewer #1 (Remarks to the Author):

Report on „Searching for new particles beyond the standard model with a single electron-spin quantum sensor“ after the first revision.

The method shown in this paper to search for scalar pseudo-scalar electron spin interaction with exotic particles is really compelling. It could open the direct searchable window to larger masses of the interacting bosons, in most cases ALPs (axion like particle). The main concerns the reviewers had raised over that paper, the missing discussion of systematic errors, is now rigorously addressed by the authors. Also minor suggestions of the reviewers are nicely incorporated into the revised version. But there is still one issue, which has to be settled before I can recommend this paper for publication.

From my experimental experience with magnetic effects caused by diamagnetism I'm in doubt about the result on the systematic effect of the diamagnetic half sphere. In order to get analytic results, I calculated the phase shift caused by a diamagnetic sphere (see attachment) on the spin echo signal. The result should be correct within a factor of two or less, because the second half sphere, I add in my calculation, has a larger distance to the NV. From this calculation I got an absolute value for the systematic phase shift caused by a diamagnetic sphere of 2.2×10^{-4} . This value is 2 orders of magnitude smaller, than the statistical error of the measurements, and therefore does not affect the result. But its value is 6 order of magnitude bigger than the result given at the supplementary information page 17 (1.7×10^{-10}). Both results can't be right, because they should agree within a factor of two. So please check Your calculations regarding the effects of diamagnetism. Further I claim that the improvement given by the dashed red line within Figure 4 is unrealistic, particularly because the improvement is based on increasing the spin echo time τ and the amplitude of vibration. Both properties increase linearly the systematic effect caused by diamagnetism (see attachment) and therefore there is not much room before the systematic uncertainty attributed to diamagnetism becomes dominant and the limiting factor.

Reviewer #2 (Remarks to the Author):

I have reviewed the revised manuscript and I am satisfied with the changes the authors have made to the manuscript. For the reasons given in my original review, it is now my opinion the the manuscript be published in Nature Communications.

Reviewer #3 (Remarks to the Author):

The authors made a big effort in addressing all the referees comments in great detail and revising the manuscript accordingly. In particular the discussion of systematic and statistical errors as well the presentation of experimental details have improved significantly. Given the positive statements by the other two referees about the significance of the present work, I do recommend the present article for publication in Nature communications in view that this work will inspire new efforts in the search for particles.

The point-by-point response to the referees' comments is shown in the following (the referees' comments in roman font and our replies in italic).

Reviewer #1 (Remarks to the Author):

Report on "Searching for new particles beyond the standard model with a single electron-spin quantum sensor" after the first revision.

The method shown in this paper to search for scalar pseudo-scalar electron spin interaction with exotic particles is really compelling. It could open the direct searchable window to larger masses of the interacting bosons, in most cases ALPs (axion like particle). The main concerns the reviewers had raised over that paper, the missing discussion of systematic errors, is now rigorously addressed by the authors. Also minor suggestions of the reviewers are nicely incorporated into the revised version. But there is still one issue, which has to be settled before I can recommend this paper for publication.

We thank referee #1 for careful reading.

From my experimental experience with magnetic effects caused by diamagnetism I'm in doubt about the result on the systematic effect of the diamagnetic half sphere. In order to get analytic results, I calculated the phase shift caused by a diamagnetic sphere (see attachment) on the spin echo signal. The result should be correct within a factor of two or less, because the second half sphere, I add in my calculation, has a larger distance to the NV. From this calculation I got an absolute value for the systematic phase shift caused by a diamagnetic sphere of 2.2×10^{-4} . This value is 2 orders of magnitude smaller, than the statistical error of the measurements, and therefore does not affect the result. But its value is 6 order of magnitude bigger than the result given at the supplementary information page 17 (1.7×10^{-10}). Both results can't be right, because they should agree within a factor of two. So please check Your calculations regarding the effects of diamagnetism.

We thank referee #1 for the calculation to check the systematic effect caused by diamagnetism. Following the referee's suggestion, we have checked again our calculations carefully and reconfirm that our calculations are correct. To ensure the validity of the result, we further calculate the magnetic field on the NV center caused by the diamagnetism of the half sphere with numerical integration. The result with numerical integration is in accordance with that calculated with analytical expression (equation S37 in the supplementary information).

To dismiss the discrepancy between Referee #1's calculations and ours, we also calculate the magnetic field caused by the diamagnetism of a sphere. For clarity, the angle θ and definitions of the unit vectors are shown in Fig. R1. The magnetization field of the sphere is calculated to be

$$\vec{B}_M(R) = -\frac{2}{3}\chi B_0(\cos\theta \vec{e}_r + \frac{1}{2}\sin\theta \vec{e}_\theta)\left(\frac{R_0}{R}\right)^3, \quad (1)$$

which agrees with the results given by Referee #1, apart from an insignificant factor of $-2/3$. Hence, the component of the magnetization field parallel to the NV symmetry axis is

$$\begin{aligned} \vec{B}_{M,\parallel}(R) \\ = -\frac{2}{3}\chi B_0(\underbrace{\cos\theta \vec{e}_r \cdot \vec{e}_{S_{NV}}}_{-\cos\theta} + \frac{1}{2}\sin\theta \underbrace{\vec{e}_\theta \cdot \vec{e}_{S_{NV}}}_{\sin\theta})\left(\frac{R_0}{R}\right)^3 \quad (2) \end{aligned}$$

Compared with the second equation in the calculation of referee #1, we notice a difference of a minus sign. The two terms in Eq. (2) cancel each other when $\theta =$

$\arccos(1/\sqrt{3})$, i.e., the magnetization field is perpendicular to the NV symmetry axis.

The phase shift caused by the diamagnetic sphere is calculated to be $2.0E-10$ rad, which is compatible with the calculated phase shift caused by the half sphere.

In the revised version, to avoid possible confusions, we add a figure in the supplementary information to clarify the angle θ as well as the direction of the magnetic field caused by the diamagnetic half sphere (see list of changes, 1).

Further I claim that the improvement given by the dashed red line within Figure 4 is unrealistic, particularly because the improvement is based on increasing the spin echo time τ and the amplitude of vibration. Both properties increase linearly the systematic effect caused by diamagnetism (see attachment) and therefore there is not much room before the systematic uncertainty attributed to diamagnetism becomes dominant and the limiting factor.

With the discrepancy between our calculations and Referee #1's estimations on the effect of diamagnetism being resolved, and as the systematic effect caused by diamagnetism has already been taken into consideration in the calculation of the improvement, our original discussions on further improvement remain to be valid.

Figure R1

Reviewer #2 (Remarks to the Author):

I have reviewed the revised manuscript and I am satisfied with the changes the authors have made to the manuscript. For the reasons given in my original review, it is now my opinion the the manuscript be published in Nature Communications.

We thank referee #2 for recommendation of publication of our manuscript.

Reviewer #3 (Remarks to the Author):

The authors made a big effort in addressing all the referees comments in great detail and revising the manuscript accordingly. In particular the discussion of systematic and statistical errors as well the presentation of experimental details have improved significantly. Given the positive statements by the other two referees about the significance of the present work, I do recommend the present article for publication in Nature communications in view that this work will inspire new efforts in the search for particles.

We thank referee #3 for recommendation of publication of our manuscript.

List of changes

- 1. Supplementary Information: A figure (Supplementary Figure 7 in the revised version), which shows the angle θ and the direction of the magnetic field caused by the diamagnetic half sphere, is added to avoid possible confusions. Corresponding description of the figure is added in Supplementary Note 4.*
- 2. Reformatting the main text and supplementary information in various places to ensure that the manuscript complies with the format requirements of Nature Communications.*

REVIEWERS' COMMENTS:

Reviewer #1 (Remarks to the Author):

Report on „Searching for new particles beyond the standard model with a single electron-spin quantum sensor“ after the second revision.

Now after the last issue I raised about systematic effects caused by diamagnetism is settled I fully recommend this paper for publication. The discrepancy on the size of this effect, was caused by a wrong sign in my calculations. I thank the authors for their clear answer concerning this issue. Also I appreciate the changes of supplementary figure 7. Now it becomes clear that the chosen field direction really helps to suppress diamagnetic false effects.

The point-by-point response to the referees' comments is shown in the following (the referees' comments in roman font and our replies in italic).

Reviewer #1 (Remarks to the Author):

Report on "Searching for new particles beyond the standard model with a single electron-spin quantum sensor" after the first revision.

Now after the last issue I raised about systematic effects caused by diamagnetism is settled I fully recommend this paper for publication. The discrepancy on the size of this effect, was caused by a wrong sign in my calculations. I thank the authors for their clear answer concerning this issue. Also I appreciate the changes of supplementary figure 7. Now it becomes clear that the chosen field direction really helps to suppress diamagnetic false effects.

We thank referee #1 for recommendation of publication of our manuscript.

List of changes

- 1. The title of the paper has been modified to 'Searching for an exotic spin-dependent interaction with a single electron-spin quantum sensor' according to editor's suggestion.*
- 2. Various other minor revisions have been made according to editor's suggestions.*